# Analytic Background in the Neuroscience of the Potential Project “Hippocrates”

**DOI:** 10.3390/brainsci13010039

**Published:** 2022-12-24

**Authors:** Irina Trofimova

**Affiliations:** Laboratory of Collective Intelligence, Department of Psychiatry and Behavioural Neurosciences, McMaster University, 92 Bowman St, Hamilton, ON L8S 2T6, Canada; iratrofimov@gmail.com

**Keywords:** constructivism, project Hippocrates, neurotransmitters, neurochemical framework FET, “Throw & Catch”, Specialized External Phenotype

## Abstract

This paper reviews the principles identified in analytic neuroscience that could be used in the setup of an international project, “Hippocrates” (H-project), named after the author of the endocrine theory of temperaments. The H-project can aim to summarize the findings in functional neurochemistry of consistent behavioural patterns (CBPs) in health (such as temperament traits) and psychopathology (symptoms of psychiatric disorders); to have systematically structured neurochemical investigations; to have an analysis of CBPs that include all ranges of behavioural histories and to have these modules complemented by regional contrasts related to climate, diets and other bio-environmental factors. The review highlights the benefits of constructivism and illustrates the contrast between constructivism and current approaches in terms of analytic and methodological aspects. (1) “Where” the neurochemical biomarkers should be measured: the review expands the range of needed measurements to out-of-brain systems, including environmental factors, and explores the concept of Specialized Extended Phenotype. (2) “What” should be measured but is missing: the review points to the need for measurement of the “Throw & Catch” neurochemical relays; behavioural and neuronal events contributing to the consistency of the CBPs but not documented in measurements. (3) Structuring the H-project’s setup: the paper briefly describes a proposed earlier neurochemical framework, Functional Ensemble of Temperament that that accommodates the neurochemical continuum between temperament and symptoms of psychiatric disorders. This framework is in line with documented “Throw & Catch” neurochemical relays and can also be used to organize data about the personal and professional history of an individual.

## 1. A Need for International Cooperation for Analysis of Neurochemical Regulation of Behaviour

### 1.1. It’s Time to Start Sorting What We Know about Neurochemical “Soups” of Behavioural Regulation 

Classification of the functional “parts” of the nervous system and identification of its neuronal biomarkers would help to predict healthy behaviour and could make more transparent treatment approaches in cases of psychopathology. For these reasons, many resources were already spent on mapping the brain and classifications of consistent behavioural patterns (CBP) [1,2]. As suggested earlier, the CBP abbreviation unifies the concepts related to consistent individual differences in behaviour, whether temperament and personality traits of healthy people, symptoms of mental disorders or semi-clinical patterns in-between on this continuum [2,3,4,5,6,7,8].

In the search for CBP biomarkers and sorting out functional partitioning of nervous systems, the structures-oriented approach, i.e., neuroimaging, has become dominant in neuroscience in the past 50 years. Several international projects were designed around the idea that structural connectivity of the grey and white matter of the brain can show all that we need to classify biomarkers of CBP. Another approach, processes-oriented (represented by neurochemistry and endocrinology), remained more in the shadow of neuroscience. Developing international neurochemistry projects is still a work in progress, despite several strong national centres conducting this type of research. The slow integration of an international project studying neurochemical biomarkers of CBPs is partially due to often invasive, resource-consuming techniques involving medical staff, animal lives, expensive chemicals, and control of multiple interactions within neurochemical systems. Indeed, technological progress, not invasive nature, fast turn-around of results, and lower cost of neuroimaging technology compared to invasive neurochemical experiments can explain the dominance of the structures-oriented approach in neuroscience. In fact, the “List of neuroscience databases” published on Wikipedia consists of 57 databases, mainly sharing neuroimaging data (MRI, CT-scan, mapping brain structures and their connectivity) [9]. None of these 57 projects uses data about the neurochemical compositions of brain areas, the concentration of neurotransmitters or receptor density, even though such works have been known for almost half a century since the fundamental database collected by Nieuwenhuys [10]. 

Meanwhile, functional activities of the brain’s structures (including dendrites and axons) largely depend on neurotransmission, i.e., a primarily chemical process. Brain structures and specific synapses vary dramatically regarding the types of neurotransmitters and receptors used during their activities. These variations are not random but reflect the functionality of neurochemical systems in behavioural regulation (as reviewed earlier in [7,8,11,12,13,14,15,16,17,18,19]. Moreover, temperaments and symptoms of psychopathology are based on the same neurochemical cycles, and the idea of a continuum between them was supported by neurochemical correlates of several temperament traits:neuroticism [20,21], low endurance [22], rigidity (as low plasticity) [23,24], impulsivity [25,26,27], emotional dispositions [16,28], (low) empathic processing [29,30], compromised probabilistic (thought) processing [31], sensation/risk seeking [32,33], and (low) sustained attention [34,35]. This validates the importance of development of an international project looking into these hidden but crucial neurochemical processes of behavioural regulation.

We suggest the name of Hippocrates as the title of this project (here abbreviated as H-project) to honour the author of the psychological concept for the neurochemistry and endocrinology of individual differences, i.e., the concept of temperament. Importantly, Hippocrates was a very experienced and knowledgeable medical doctor, credited as the father of the modern systematic approach to medicine. His concept of “*temperamentum*” (that means “mixture” in Latin), emphasizes not only the chemical nature of temperament but also the role of bodily fluids in behavioural regulation, supporting the principle of embodiment in behavioural regulation. The main aims of the H-project can be the mapping of neurochemical biomarkers of CBPs and the facilitation of the development of neurochemistry-routed taxonomies of CBP (i.e., classifications of temperament traits in healthy and symptoms of psychopathology in psychiatric cases). This project should include experimental, educational and also analytic parts. 

However, the setup of the H-project faces another problem: the extreme complexity, diversity, transience and multiplicity of neurochemical processes.

### 1.2. Complexity of Neurochemical Systems Require Theories of Their Regulatory Principles beyond Excitation-Inhibition

The task of sorting out the functionality of neurotransmitters (NTs) appeared to be far from trivial [10,36]. More than a dozen of non-peptide and over 100 peptide neurotransmitters have been identified, each having different functionality and distribution in the nervous system. In addition, many of these neurotransmitters have a diversity of receptors, each having different functionality and location pattern. For example, there are (so far discovered) 5 types of dopamine (DA) receptor, 9 types of adrenergic receptor, 14 + types of serotonin (5-HT) receptor, two families of acetylcholine (ACh) receptor (each having 5–12 subtypes). A similar diversity of receptors has been found for histamine, Gamma-Amino-Butyric Acid (GABA), glutamate (Glu) and endogenous opioids [36]. Complexity of neurochemical biomarkers of the CBPs does not end here. Neurotransmitters regulate one another’s activity in a contingent manner via several mechanisms with different release patterns and different mediators depending on the intensity of stimulation and the location and density of receptors. The same two NTs can be rivals under one condition (or location), suppressing each other’s release, or partners under another condition/location, having a co-release. Receptor complexes often use heteromer arrangement where two neurotransmitter systems have joined receptors locally regulating each other’s release [37,38,39,40]. Moreover, neurotransmitters often use so-called “volume transmission”, i.e., transmission in which their NT is released to the extracellular space and acts on whatever neurons are there receptive to it [38,39,41,42]. Finally, neurochemistry is not only the discipline studying the “soups” that regulate human behaviour. A proper analysis of neurochemical biomarkers of CBPs should be multidisciplinary, including the control of variables studied in endocrinology, gut psychiatry, psychopharmacology, neuroimmunology and biological psychiatry [2].

For efficient integration of existing and possible new data regarding the complex and closely entangled neurochemical biomarkers of CBPs, a reliance on blind statistical data mining is insufficient. There should be underlying competing theories and verifiable hypotheses that organize the data collection and analysis. After all, analytical work is part of science, monitoring what was done and what is missing. 

This paper reviews several rarely discussed methodological and analytical considerations that, in our opinion, should be taken into account in the design of the H-project. These considerations stem from the constructivism paradigm. After the introduction to constructivism with examples of the pro-active nature of neuronal regulation, the paper illustrates the contrast between constructivism and current approaches in terms of: (1)“Where” the neurochemical biomarkers should be measured: the review expands the range of needed measurements to out-of-brain systems, including environmental factors, and explores the concept of Specialized Extended Phenotype.(2)“What” should be measured but is missing: the review points to the need of measurement of the “Throw & Catch” neurochemical relays; behavioural and neuronal events contributing to the consistency of the CBPs but not documented in measurements.(3)Structuring the setup: the review briefly describes a proposed earlier neurochemical framework that accommodates the neurochemical continuum between temperament and symptoms of psychiatric disorders. This framework is in line with documented “Throw & Catch” neurochemical relays and can also be used to organize data about the personal and professional history of an individual.

## 2. Introduction to Constructivism

### 2.1. Bernstein: “Repetition without Repetition” and Constructivism

*Constructivism* suggests that behaviour and all its psychological elements are not reactive, being evoked by stimuli ondemand but generated (constructed) every time afresh in somewhat similar or different configurations [43,44,45,46,47]. The principle that psychological phenomena have a constructive nature was first experimentally demonstrated using original tracking devices in the work of the Russian psychologist Nikolay Bernstein in the 1930s (literally creating a new discipline of kinesiology) [43,44,45,46]. English cognitive psychologist Frederic Bartlett [48] also proposed in the mid-1930s that memory has a constructive nature. This constructivism principle was described in a range of bio-behavioural sciences (kinesiology, psychophysiology, cognitive, developmental, ecological, educational psychology, psychological modelling and psychology of emotions, see [7,8,49] for reviews). At the neuronal level of behavioural regulation, it has been shown that brain connectivity is very plastic and that neuronal ensembles of the brain re-organize themselves with a change in situational contexts and tasks [50,51,52]. 

The generative nature of psychological processes is getting a gradual acceptance in modern science; however, some scientists are sceptical about Bernstein’sinsight that *all processes and actions are generated every time anew based on available resources, needs and past experience*. This fundamentally important insight is summarized by Bernstein as the “*repetition without repetition*” principle and applies to the generation of even simple actions ([44], pp. 204, 436). This contrasts with the conditioning theory and behaviourism principles stemming from the work of another, much more influential Russian psychophysiologist Ivan Pavlov. These were precisely the years when Pavlov’s work just gained enormous popularity not only in Russia but around the world. In fact, Bernstein’s fundamental book “On the construction of actions” was fully prepared for publication in 1936 but was pulled off print at the last moment because Ivan Pavlov died the same year [44]. 

Interestingly, Pavlov and Bernstein had a good working relationship. Pavlov agreed with Bernstein’s ideas on constructivism and did not approve simplification of his work on conditioning in the form of American behaviourism, pointing to a range of biologically based and not experience-based individual differences (temperament). Meanwhile, the suggestion that every action, even a repeated one, is constructed anew was confronted with much criticism in Russian academia. Bernstein’s work was not known to the West till the mid-1950s; he eventually lost his academic position and the support of many colleagues. To be fair, even experienced psychologists and neuroscientists still have a hard time believing that identically looking actions, such as repeated hammering, are not just inductions of established brain wiring that resulted in a conditioned reflex after learning the action. It is important, therefore, to continue discussions on whether or not the mechanisms in brain functioning support the constructivism paradigm.

Meanwhile, studies in neuroscience have described multiple phenomena confirming the constructivism principle, some of which will be described below. 

### 2.2. Neurotransmission under a Magnifying Glass: Between Compositions and Decompositions

Young neuroscientists are often taken by the fact that synaptic wiring increases with experience. The Connectome-like projects [9] represent the human brain as an analogy to radio circuits permanently wired between the brain structures that each have a certain function. In this model, it is a matter of tracing this wiring and mapping the brain’s functional parts activated during specific tasks and contexts.

This is not what happens at the receptor (i.e., molecular) level. Most neurotransmitters (NT) and neuropeptides (NP) are not stored in synapses, ready for release. Instead are synthesized and immediately decomposed after use, with different mechanisms related to these cycles for each NT. NT release is also not a simple “knock → gates open” process. It proceeds in several stages, and each stage involves a cascade of contingent transformations regulated by mediators such as GABA, Glutamate, G-protein coupled receptors, transcription and neurotrophic factors, enzymes, metabolites, ATP, calcium and other chemical systems, as well as the regulatory impact of NTs and NPs on each other [36]. Each of these mediators has the capacity to disrupt or change the pace of neurotransmission. A similar complexity and contingent construction were described in the action of opioid receptors (OR) [53,54]. After transmission, the molecules of NTs are quickly metabolized at their releasing sites and, therefore, must be generated all over again later.

Moreover, the representation of the brain as wires transmitting environmental stimuli and highlighting their adverse or rewarding value does not explain why the most common type of transmission is not the classic (fast) ion-ligand synapse but rather the G-protein-coupling receptors (often slow and sloppy) mechanism (GPCR). The GPCR have a variable speed of response, between an immediate effect and a few days, as it requires multiple protein mediators. The GPCR transmission also often is not exact but rather fuzzy in action (see [55] for details). This is not something that would be expected if the receptors should mechanically pass a signal as fast as possible. Yet, the GPCR mechanism dominates the types of neuromodulators’ receptors: six out of seven classes of 5-HT receptors, all DA receptors, more than half of NE receptors and half of ACh receptors are GPCR-based [36]. 

All this means that the generation of neurotransmission in every single synapse on the 10^10^ brain cells is not an automatic but rather a contingent process. During the construction and disassembling of NTs in the brain and gut, the availability of the chemical substrates, as well as variations in composition, can be affected by genetics, environmental factors, physiological state of the body, state of the supporting microglia cells and the state of the brain cells themselves that manufacture the needed components. 

### 2.3. “Multiplicity of Candidates” Supports “Repetition without Repetition” at Many Levels of Behavioural Regulation but Leads to the Degrees of Freedom Problem

One of the principles that ensure the “repetition without repetition” mechanism is the “*multiplicity of candidates*”, i.e., the existence of several elements with similar functionality that could be used for a specific function. The best analogy would be having different orchestras or music groups playing the same melody but having different performing members, even though the melody stays the same. Similarly, there are multiple neurons within each brain structure and so the same action could be repeated using different representatives of specific structures but literally constructed anew since these representatives are different even though the involved structures might be the same. After all, for every action, even stereotypic and automatic, there is a multi-stage selection process, selecting and “hiring” performing neurons out of multiple ready candidates. There are at least ten times more “offers” (outputs from neurons to other neurons offering a pulse) than “hires” (accepted inputs to the neurons) in the cortex, illustrating the “multiplicity of candidates” principle [56,57,58]. Neurons do not always fire after being excited by the inputs or respond to the received pulses immediately: responses from dendrites are being gathered, first generating a dendritic wave of electrical current that flows to the cell body. Axons also cannot generate another pulse immediately (due to the recovery period) [57,58,59]. 

A similar “multiplicity of candidates” principle is common in many aspects of nervous activity, behaviour in general and the functioning of natural systems (Figure 1). There are multiple, and not point-targeted, release of neurotransmitters to reach multiple receptors related to one function; there are multiple forms of neurotransmission within each neurotransmitter system; there is overlap in functionality between neurochemical systems, etc. (see more examples in classic handbooks on neuroscience [36,60]); there are multiple ways for a human to grab an object. The “multiplicity of candidates” principle is seen in the generation of“multiples of everything”, where only some offers, and not necessarily the best ones, will be accepted. Yet, this principle is behind the fact that we can have repeated actions of very similar content even though neurons that performed early repetitions have not recovered yet, and different neurons from the same brain areas are involved. In other words, having multiple productions of actions within neuronal or behavioural levels using similar components and similar dynamics makes these unique constructions look like repeats of the same action.

Moreover, Bernstein showed that the generation (“construction”) of action proceeds through the integration of behavioural alternatives selected and sequenced for their relevance at many levels of behavioural regulation [43,44]. Having these levels explains the “repetition without repetition” idea of Bernstein since, indeed, identical actions could be controlled by different levels (Bernstein identified five such “levels of control”). Bernstein was the first to experimentally demonstrate that first-trial actions have maximal conscious control and pointed to the role of the cortex in its regulation. Then, as he showed in the experiments on seemingly stereotypic actions such as hammering, walking, typing, piano playing, etc., with learning, the control over the action construction is passed from the upper levels (i.e., more conscious integration) to the lower levels (more automatic integration). Yet, when the construction of action faces challenges, it shifts back from automatic to more conscious levels. Bernstein, with the assistance of the famous neuropsychologist Luria, who shared some clinical material for Bernstein’s book, identified the cortical—ventral striatum networks as more conscious levels of control over the construction of action, whereas more automatic levels were assigned to thalamic-globus pallidum (GP) and cerebellum networks [43].

This insight was first confirmed in psychology, with descriptions of several levels of control in cognitive and motor regulation starting from the work of Treisman on perception [61]. Then, it was validated in neuroscience, by the evidence of differential involvement of ventral and dorsal complexes in basal ganglia during the transitions between novel/uncertain and learned actions [62,63,64,65,66,67]. This work has demonstrated that with more determination of the program of actions (with habit formation and more automatic integration), control over the integration of action is passed from the cortex to the ventral striatum (consisting of the nucleus accumbens, part of the olfactory tubercle and the ventromedial part of caudate) to the dorsal striatum (i.e., the main caudate nucleus and putamen) [67,68]. 

Conversely, with an increase in the complexity of the task, control over integration is passed back to ventral-striatal-cortical networks. When a new program of actions is required due to the novelty or complexity of a situation, and there is no ready script of actions, only then the full power of the cortex and hippocampus is engaged. Otherwise, current changes in the situation often rely on the ventral striatum-cortical monitoring and moderate adjustment of actions that this network constructs and passes to the dorsal striatum and cerebellum for final choices of degrees of freedom.

Relevant to understanding the constructive nature of behaviour, it is important to underline that Bernstein was also the first scientist who outlined and experimentally demonstrated “the *Degrees of Freedom problem*”. This problem since then has been well-recognized in neuro-behavioural sciences and cybernetics.

The DF problem relates to the fact that there are simply too many stimuli around that our nervous systems can react to, and there are also multiple ways to “react” in every behavioural act. Out of all possibilities, it is a great challenge for the nervous systems to select what to be attentive to. Plus, since the nervous system has to suppress the majority of stimuli(Ss) and actions (Rs), it would be easy to miss something important if such suppression would be “by the bulk” and not selective. Moreover, the nervous system has to decide to what degree the action should involve novel elements and make this decision fast and, preferably, in advance. The multiplicity of *Ss* and *Rs* and their constant selection-out by the nervous systems means that in a majority of cases, behaviourdoes not follow presented stimuli (and, therefore, does not follow the *S → R* model) but rather suppresses them.

The “degrees of freedom” concept was well-adopted in cybernetics and statistical mechanics, especially in the explanation of the concept of “entropy” [69,70]. “Action in multiples”, or collective modes in actions and information processing, are common in nature, allowing colonies of “collective” animals to perform what is considered “collective intelligence” tasks (such as basic calculations, estimations of the probability of events and building very sophisticated infrastructures) [71,72,73]. Neuronal mass action became a subject of mathematical modelling, with the most famous model, “Perceptron”, offered initially by Frank Rosenblatt in 1943 and then developed by Minsky and Papert in the 1970s [74]. 

Action in multiples or mass action is also seen in neurodynamics. Walter Freeman, who conducted one of the most comprehensive works on mass action in neurodynamics pointed out that there are thousands of inputs to a single neuron in the cortex that “vote” with their contributions, and this vote results in the wave coming from this neuron, eventually either transforming or not transforming to a pulse [75]. The mass action of neurons, therefore, produces a filter of information, in which there are multiple “votes” pro and against transmission of the pulse coming to each neuron, and so the selection and the processing of information proceed simultaneously by multiple selection-voting stations. Therefore, the postsynaptic potential’s strength decreases with the distance between the synapse and the cell body. Hence, as Freeman points out, the contribution from a distant synapse is weaker than that from a nearby synapse (later confirmed in multiple studies in neuronal transmission) [76]. This means that the simplified presentation of synaptic transmission as a guaranteed “transfer of the information” is wrong: a single transmission does not help information processing because it depends on other synapses and the locations of the synapse.

### 2.4. Anticipatory Neurodynamics Illustrates the Constructivism Paradigm

Freeman’s experiments and mathematical analysis using nonlinear dynamics revealed more complex, contingent and structured dynamical patterns than just “voting for or against” the spread of neuronal pulses. Freeman described the presence of several competing dynamical patterns (“wings of chaotic attractors”), generated internally by the neural networks [56,59,75,77,78,79]. Neurodynamics can follow any of these patterns, and switching between them remains unpredictable until the incoming information brings some certainty to these conflicting potentials. Freeman underlined that these anticipatory patterns are not shaped by the stimuli directly but by previous experience with those stimuli; they are “constructions by brains, not merely read-outs of fixed action patterns” ([75], p. 83). The background neurodynamics provides the system with continued “open-endedness”, making it ready to respond to information without the requirement for an exhaustive memory search [75,78]. This anticipatory readiness for several behavioural alternatives significantly speeds up the future selection of degrees of freedom. 

The anticipatory nature of perception and cognition was noted in psychology around the time of Freeman’s experiments (for example, see [80]); however, these experiments brought the evidence from neuroscience regarding anticipatory nature of orientation. Interestingly, Bernstein also wrote extensively on the idea that the nervous system generates a “model of the future” while constructing the action and choosing the degrees of freedom. This model of the future facilitates the selection of environmental alternatives to be attentive to [43,44,46]. Freeman’s neurodynamics validates the constructivism principle as ithighlights the pro-active nature ofselection of DFs within the nervous system.

## 3. Where to Measure: Not Just in the Brain

### 3.1. Back to Hippocrates?

Neurochemical investigations of the brain should be an essential part of the H-project. However, neurochemical regulation of human and animal behaviour involves the systems outside the brain. Human bodies are open and dissipative systems, and all neurochemical systems compose and decompose their components using external sources of nutrients, sunlight, physical and social contacts and information. Modern findings of hormonal regulation involving endocrine glands and the emerging discipline of “gut psychiatry” describing the contribution of microbiota to the regulation of the HPA axis fully support this idea [81,82,83,84]. Gut psychiatry and endocrinology, therefore, are essential disciplines that should be part of the H-project. 

An important part of the H-project should be investigations of the interaction of neurochemical regulation of behaviour with diets. Multi-cellular animals or humans prime their microbiotas and nervous systems to certain diets during the first stages of their ontogenesis. Then, in line with the SEP concept, their nervous systems (under the influence of their microbiotic and metabolic needs) construct their behaviour and build environmental infrastructures to ensure the resources of these particular diets. Phenotypes that belong to the same biological species can eventually diverge to different functional groups just because these phenotypes are consistently exposed to different diets. 

One of the examples of a possible influence of diets on CBPs relates to tryptophan, a necessary component for 5-HT composition. Even though 5-HT regulates many body and brain systems, neither body nor the brain can manufacture their own tryptophan, so they use two sources to get it: diets and microbiota. Considering the essential role of serotonin disregulation in psychiatric disorders, it is reasonable to suggest that contrasts in diets related to tryptophan can be associated with differences in these disorders or at least national differences in some CBPs. The impact of the tryptophan diet was, in fact, reported in association with depression in clients with neurodegenerative disease [85,86] or animal models of chronic unpredictable stress (Wang et al., 2022). As another example, soy products and the estrogen-like properties of their component, isoflavone, were noted to have an impact factor on the activity of organs that depend on estrogen, including the hypothalamus/pituitary, the other brain regions and the uterus [87]. Interaction of estrogen with all leading neurotransmitter-managers (5-HT, DA, NE and ACh) can explain findings that isoflavone-rich food may improve overall mood and behaviour [88,89], including depression [90]. Other examples of dietary factors of CBP regulation include the impacts of iodine [91], chocolate [92], caffeine [93] and vitamin B [94] on anxiety and depression. 

Microbiota is the first major composer of the components that eventually is used in neurotransmitters’ composition, soit should be a subject of the H-project contrasting various regional samples for specific biota species. The next stage of the composition of neurochemical components uses the liver as a manufacturer of enzymes needed for neurotransmitter synthesis. Therefore, it is essential that neurochemical investigations of brain systems should be complemented by investigations of both, microbiota and liver enzymes for each regional sample. 

Moreover, the use of common psychostimulants in diets varies between regions and countries, so the setup of the H-project can include a matrix contrasting the samples drawn from these regions. Examples include the samples with a high (France, Italy, Russia) and low (China, Japan, New Zealand) use of nicotine; high (Sweden, Norway, UK) and low (Iran, India, some African countries) use of alcohol; and high (Iran, Iraq, Turkey, Afghanistan) and low (Switzerland, Japan, Ukraine) use of opioids.

Moreover, other “non-brain” psychophysiological and biological processes—physical touch [95,96], changes in hormones during perceived social support [97,98,99], diets [100], exposure to natural green environments [101,102]—were reported as making significant contributions to CBPs. However, in many cases, the “given by birth” microbiota, immune challenges, amount of sunlight, green environments, and physical touch environments are still not good enough for an individual’s needs. This explains why many people prefer to move and live in other regions when financial or employment factors do not dictate their decisions. When they relocate, however, in most cases, people’s choice of diets stays the same even though it is often difficult to find proper ingredients for their diets in new locations (as seen in immigrants’ communities).

Consistency of diets illustrates the importance of individual microbiota for psychological and physical functioning. Once microbiota populates a child’s guts at the beginning of life, it normally keeps its individual profile, making it difficult for a grown-up individual to change diet preferences after relocation to another region [103,104]. There is, therefore, many “outside-the-brain” factors affecting the choice of people’s CBPs. Pets’ owners know similar consistency in diet preferences of their pets: change in offered foods can make animals agitated, aggressive, or depressed, and they often refuse to eat new food.

### 3.2. Back to Empedocles? Regional Contrasts in Environmental Factors Relevant to the H-Project

Investigations into neurochemical biomarkers of CBPs should include the interaction of neurochemical systems with environmental factors influencing these systems. This brings us to the ancient theory of Empedocles, highlighting the role of seasons and “body fluids”, including the fluids processing nutrients. As discussed above, the roles of microbiota, immune systems, amount of sunlight, exposure to infections, types of diets, physical touch and cultural customs for socialization validate these ancient scholars’ ideas. Biological factors of behavioural regulation do not amount to the brain or even to the brain+body system; they depend on the physical SEPs that support this system. In this sense, the Connectome-like projects miss a big hank of biomarkers of CBPs. To have scientifically verified maps of the roles and interactions of environmental and body-brain biomarkers of the CBPs, the H-project might benefit from contrasting the cultural and regional samples that have high and low expression of these environmental factors. For example, the samples could be contrasted by the high (Italian, Cuban, Turkish communities) and low amount of sunlight (Scandinavian or Northern Canadian communities); high (South American countries, Italy, Arab communities) and low (Scandinavian countries, Germany, UK) amount of culturally acceptable physical touch; high (China, India) and low (Canada, Hungary, Sweden) population densities that might be a factor in immune challenges.

These regional comparisons are possible only with the cooperation of multiple scientific and public communities representing various counties. Therefore, similarly to the CERN project in high-energy physics, the H-project could be developed only with a setup of an international Task Force and extensive discussions at many levels of scientific and international organizations. Academic discussions on this subject have already started [2].

Last and not least is the mandatory contrast related to sex differences, which, as discussed elsewhere, should move beyond the classic sex dichotomy to more objective measures of gonadal hormones [2,105].

### 3.3. T&C Specialized Extended Phenotypes (Behavioural Bubbles) as a Useful Concept for CBP Research

The role of the environment in behavioural regulation is widely discussed, but the setups of international projects struggle with the diversity of aspects in individual-environment interactions. Earlier, we pointed out the benefits of going beyond seeing an individual as a passive “taker” of environmental influences, even if this taker is selective. The environment-individual interactions, in our opinion, have a two-way road, as highlighted in Richard Dawkins’ concept of *extended phenotype* [106]. Animals and humans actively build and select elements of shelters, routines related to food sources, and social relations that later determine and organize their choice of actions [107]. Individuals use tools, sets of relationships and accommodations they have developed before. Moreover, many cognitive processes depend on the way how individuals use their environments to store their previous experiences for future use. Different animals used different “tricks” to their environment to keep their decision-making elements for future use (such as leaving marks on the trees, spraying, establishing mating partnerships, building dams and nests, storing nuts and even meat). The most dramatic examples are humans, whose extended phenotype was so extended that their bodies at birth evolved to be totally dependent on that phenotype, even for the ability to suck mother’s breasts or keep their bodies warm. 

Importantly, these made-up infrastructures are still the product of the selection of DFs. They exist outside their bodies and nervous systems but become major (self-chosen) regulators of their behaviour, indeed supplementing the phenotypic behaviour. In this sense, the integrations of actions only partially depend on the composition of brain neuronal ensembles and partially—on the physical and social infrastructures they voluntarily created outside their organisms during their lives. The evolution of nervous systems in tune with such “external infrastructure” was summarized not only by Dawkins but also by the authors highlighting the perspectives of ecological psychology [108].

The concept of extended phenotype was further developed into the idea *of Specialized Extended Phenotypes* (SEP) [7,19] and diagonal evolution [7,49,109]. SEP theory acknowledges that there are big differences in how people use environmental resources and regulators. When there is equal access to various technologies, resources and activities, people use only a small portion of this access, primarily compatible with their bio-behavioural abilities and needs. For example, some people like reading and tune their life to what they’ve read. Some people do not like reading but like physical activities and camping, and stuff their house with camping equipment, reminding them to plan their next camping trip. Some people are empathic to animals, and once they obtain a pet, this pet regulates their lives. Each of these types has individually shaped functional cycles, socio-ecological “bubbles” formed by their bio-behavioural capacities. These personal SEP-bubbles include individually preferred settings, objects, partners, friends, social and professional networks as their regulators, compatible with their individual differences. 

As a two-way process, the participation of people in specialized services and networks contributes to the evolution and reinforcement of these functional bubbles, with specialized infrastructures supporting the specific needs and abilities of each type of person. Following our examples in the previous paragraph, big industries developed based on the support of readers, campers and pet lovers. This two-way mutual selection and reinforcement influences between the types of neurobiological capacities and the environmental infrastructure generate segregated functional cycles, “functional bubbles”, defining the routines in which an individual is most involved. This is similar to how animals first build their supportive infrastructure and then depend on its configuration [107].

Environmental SEP infrastructures involve not only societal and informational establishments but also physical factors, such as diets and climates. Modern gut psychiatry, endocrinology and neuroimmunology have demonstrated that gut microbiota [81,82,83,84,110], immune [111,112,113,114] and hormonal systems [18,115,116] and amount of sunlight [15,117,118] impact many aspects of behaviour, with emotional regulation as the most explicit example. For example, sunlight stimulates the release of hypothalamic orexins that regulate behavioural arousal [15]. This explains the high rates of seasonal depression in Northern countries that are known for the shorter sunlight daily amounts [117]. Another example of the impact of infection on brain functioning and, therefore, the CBPs was clearly demonstrated by the recent COVID-19 pandemic [114]. 

## 4. What to Measure: The Chain of Construction or Its Isolated Spots?

### 4.1. Cycles, Run-Aways and Start-Ups Processes in Neurophysiology and Behaviour

The NTs and hormones regulating human behaviour also regulate each other’s release, composition, storage, and decomposition. A consistent under- or over-production of some components leads to specific variations in neurochemical regulation. This supports Hippocrates-Galen’s idea that dysbalances within (neuro)chemical regulatory cycles can lead to consistent biobehavioural differences, including psychopathology. Physiology is full of examples of chemical cycles swinging with healthy-range imbalances, such as oscillations between hunger and satiation; thirst and having a full bladder; feeling lonely and feeling overwhelmed with socialization; feeling informational deprivation and feeling tired from learning. Healthy neurochemical functioning involves moderate “swings”, from out-of-balance deviations in these productions (as, for example, in the case of serotonin reuptake during transmission within serotonergic receptors). Extremes in such out-of-balance states are linked with a number of psychiatric disorders with serotonin’s role in depression and opioid receptors in bipolar disorder or addiction as prominent examples. 

The processes driving both neuronal biomarkers and consistent behavioural patterns can be presented, therefore, as a chain of swings between inter-connected pendulums that trigger, amplify or suppress each other. What can possibly go wrong in contingent interactions between multiple neurochemical systems and their dependence on body’s physiology and physical environmental factors? Everything! There are many neurochemical processes that could be considered “tried and failed” compositions or processes that are too variable to be sustained. Multiple neurotransmissions do not end in prolongation of the signal, and so the signal that is passed by one cell “dies” soon after being passed to another, as described in Freeman’s neurodynamics. Similar *run-away processes* (we can use run-away as a noun here) are noted at the behavioural and even at the higher organizational levels (groups, economic and societal structures). Many people’s intentions and promises do not end in promised actions; many times, special training and experience do not lead to employment where the learned skills are needed; many times, we start sentences without finishing them but instead replace them with more adequate sentences.

For a successful composition that can pass run-aways stage and sustain as a dynamic or structural contribution to the future process, some regulatory mechanisms should trim the variance towards compatibility with supportive components of a potential cycle. The concept of a *cycle*, therefore, refers to the established multi-component processes that support sustainability of most compatible components of the cycle and suppress excessive variability that could cause a run-away of the cycle. This formalism is appropriate for the contingent composition-decomposition processes in neurotransmission and even the selective reinforcement nature of the individual-environment interaction. The establishment of “synergetic” cycles during learning of physical actions at several levels of behavioural regulation was also described by Bernstein [46,119] (pp. 174, 357–359).

There are, however, multiple successful but transient, one-time-only neuroregulatory and behavioural compositions reflected in Bernstein’s “repetition without repetition” principle. In fact, at the behavioural level, many things in life we do only once. At the neuronal level, there is significant synaptic plasticity, with synapses appearing and disappearing. The principle of mass actions makes a single association between two given cells practically insignificant for the actions that follow, as the same cells would have variability in other synapses or their chemical status but the time of action. If we ever do something again, our orientation system processes it differently, so it could be considered a different action and not a repetition. For example, we travel to some places only once; we meet many people only once; we read books and papers often only once, and we often marry only once. These one-timers successful compositions cannot be seen as cycles; instead, we can call them *start-ups.* The concept of a cycle, therefore, should be complemented by formalisms describing multiple failed compositions (run-aways) and successful but not cycled compositions (start-ups). 

The “multiplicity of candidates” principle can be interpreted as the multiplicity of “run-aways” at neuronal and behavioural levels. The fundamental benefits of this looks-like “wasteful” pluralism relate to the sustainability of cycles even though the majority of run-aways were not included in the cycles. There are always variations of the components of the natural cycles due to the generative nature of these components and its dependence on other generative factors adding variance to the compositions. There are multiple failed compositions since nothing in biology is durable and guaranteed. In this context, having a multiplicity of “siblings”, capable of doing the same job provides a fast replacement for failed components and sustainability of cycles. Moreover, the diversity of run-aways (candidates that were not involved in the cycles but are still being produced) makes cycles adaptive in case if the environmental changes require a slight change, an adjustment to the cycle (and so use of a slightly different ensemble of components) or even a completely different composition. In this context, the sustainability of the cycles supporting the periodic and consistent existence of certain configurations might not result from just one lucky configuration but is the product of multi-system cooperation. 

For example, at behavioural level, there are multiple run-away trials and projects that were started but never finished. Moreover, multiple elements in behaviour are not aligned with our plans and goals andare often distracting, i.e., anti-goal directed. Additionally, very often, people do not have a well-defined image of the final target of their behaviour but just “expose themselves to the flow”. Examples include staying in useless chatting, waiting, resting, and watching TV programs that an individual does not even like but watches because of a lack of energy to reach the remote and look for something else. The examples at the neuronal level are associated with the neuronal mass action described above, when multiple neuronal loops emerge and disappear as they were not prolonged by the waves generated pulses between neurons [59].

There are, therefore, benefits in having three types of processes within the neuronal and behavioural regulation: continuing generation of diverse run-aways most of which are not compatible with current cycles;continuing review of compatibility between generated and (ever-changing) needed DFs, tuned to the needs and capacities of the individual;continuing selection that supports existing cycles and protects them from being taken apart by the variance coming from run-aways.

A large component of the H-project can be the analysis of biographies of people, complemented by neurochemical investigations. During this biographic analysis, the formalisms of constructivism can be very useful in structuring very diverse data. 

The concept of “run-aways” behavioural and neuronal products is useful in the analysis of psychological capacities if we want to predict and understand an individual’s CBPs, whether or not they cross a clinical threshold of functionality. At the behavioural level, it would be beneficial for the H-project to collect information about an individual’s unrealized hopes and unfinished projects in the past and formally analyse these actions in terms of 12 functional aspects of behaviour identified in the neurochemical framework described below. For example, one individual started and then dropped participating in athletics, whereas another often starts and drops personal relationships. If to apply the FET framework, the first individual’s run-aways will relate to physical aspects of behaviour (indicative of healthy hypothalamic neuropeptide systems), whereas the second individual’s run-aways are indicative of attachment disorder (indicative of possibly problematic mu-opioid receptor and oxytocin systems). At the neuronal level, such processes are seen in a density of local metabolic and electrical loops in intra- and inter-cellular activity that seemingly do not contribute to information transfer. Yet, the diversity and activity of these loops can be a potential biomarker for specific healthy and clinical CBPs.

Another source of information about people’s biologically based abilities could be taken from their “start-ups”, i.e., actions or projects that they once completed but never repeated. Many actions animals and humans do only once. So establishing a wirednetworks (following the Hebbian rule “fire together → wire together”) does not make sense if the associations will never be used in the future. The focus mainly on brain structures and their connectivity in Connectome-like projects, therefore, underestimates the transience of both neuronal and behavioural processes. Moreover, as noted, a significant portion of neurotransmission happens in extra-cellular space, i.e., outside of “wired” networks [38,39,40,41,42,120,121]. If the “singletons” products of behavioural and neuronal activity are so numerous, then they should be the subject of the formal assessment. Documenting biographic details related to people’s actions outside their main professional and personal activities could be done using the formal FET structure that categorizes the actions in terms of orientational, integrational, energetic and emotionality components, as described above.

One more source of diagnostic information can be the absence of start-ups and runaways of specific nature, also classified using the 12-component framework outlined below. The absence or a low number of attempts for certain types of activities indicate biologically based weaknesses in capacities associated with CBP regulation.

### 4.2. “Throw & Catch” Relays Highlight Pro-Active and Constructive Nature of Neuronal Regulation of Behaviour

Constructivism proposes that behavioural regulation does not start from stimuli. Instead, the nervous system depends on the state of the body, which calls for the assistance of the nervous system to get metabolic, mating and accommodation resources. These calls for assistance influence pituitary-hypothalamic activity and the endocrine system and emerge as releases of relevant neuropeptides and hormones to regulate the brain and the ANS systems. Therefore, the response of specific brain structures to specific elements of events will be correlated with these elements only as long as an individual’s needs and capacities consider these elements relevant. As noted in earlier reviews [8], the neuronal infrastructure for the selection and filtering of DFs is well-known, including: modular connectivity, when projections between two brain structures go in a segregated manner [62,63,64,65,66,67,122,123]; functional partitioning within brain structures, with the most relevant example of the striatum [124] and heteromeric complexes composed of receptors related to different neurotransmitters (“receptor mosaic”) [38,39,40].

The pro-active nature of neuronal regulation of behaviour does not end with a pro-needs and pro-capacities selection of DFs in behaviour. The nervous system uses not only sophisticated mechanisms for the selection of DFs (compatible with entropy reduction concepts) but also mechanisms to self-generate DFs (not compatible with entropy reduction). This mechanism, named “*Throw & Catch*” (T&C) }, is similar to cave bats generating ultrasound waves and using sensors to process the returning profiles of these waves to “see” the cave’s surface in the darkness. In terms of the nervous system, the “T&C” principle relates to the ability of this system to have a two-way self-induced processes: the first (“Throw”) process self-generates excesses of elements (a local, internally controlled increase in their variance) that can interact with both environmental and internal systems. The second, much more structured “Catch” processes emerge in biased, strategic placement of reception sites and a diversity of selective multi-filtering mechanisms that are tuned to the needs and capacities of the individual (Figure 2B).

The “Throw & Catch” principle aligns with the natural selection theory. Cairns-Smith gave an excellent and well-illustrated description of the generation of highly complex configurations in evolution based on just two trends: “collectors” (generating the pools of configurations) and “selectors” (multiple factors of natural selection) [125]. The T&C could also be seen in the proposed earlier explanation of sex differences in phenotypic distributions, including psychological abilities and disabilities [126]. In fact, the Evolutionary Theory of Sex highlights male variability as the core property of sex reproduction mechanisms that partition phenotypes on variable and conservative partitions [126].

There are two crucial differences between Throw and Catch concepts in athletics and the T&C principle in neuronal regulation. First, unlike in football or baseball, playing with a single item to throw, the nervous system uses a spray of items that could overwhelm a non-prepared catcher. Second, similarly to the games, there is a relay that passes items from one catcher to another one. The difference is that every Catch mechanism in the nervous system needs to sort through multiple items thrown to it and select a smaller set of items to pass to the next catcher for further selection.Having sensors catching their self-generated waves or chemicals does not complete the T&C. The essential part of the Catch mechanism here is the strategic position of sensors and their selectivity, i.e., tuning to specific, subjectively chosen and not all available information. Consider an analogy with a shopper facing a supermarket’s variety, i.e., process all information coming to the n.s. As per the T&C principle, instead of asking “what you’ve got”, the shoppers, i.e., our nervous systems, do not waste time on sorting all the items in the “store of reality” but ask the environment about specific features that would satisfy their needs and ignore the majority of other environmental offers. The beauty of this principle is that nervous systems should evolve the selectivity of only one, the second (Catch) stage of this mechanism and can be not very picky and selective at the first (Throw) stage.

At the level of neurodynamics, the Throw process can be seen in the generation of multiple attractors that run in parallel, involving similar neuronal loops. This provides the “multiplicity of candidates” to be subjected to the selection by the multi-stage Catch systems of pulse generators. During this process, the selection runsseveral preferable sets of DFs used for possible multiple acting scenarios in parallel (Plan A, Plan B, Plan C and a “No-way plan”, depicted as letters A, B, “?” and a stop sign within the boxes in Figure 1 and Figure 2B). 

At the level of neurochemistry, the functional and structural differences between neurotransmitter systems give multiple examples of T&C: 

Volume transmission. Many “talks” between neurochemical systems happen not in synaptic transmissions between neurons but in extracellular space, i.e., as “volume transmission” [120,121]. Evidently, all cortical monoamines [38,39,40], acetylcholine [41,42] and likely other NT systems such as glutamate and GABA [41,127] use volume transmission by releasing NT-s to the extracellular space and being influenced by other NT-s released there as well. Similarly, volume transmission was reported within opioid receptor systems [128]. Some researchers suggest that most of the intracortical ACh [41,42] and most of the nigrostriatal DA [37] are not released at synaptic contacts but rather diffusely into the extracellular space via volume transmission. If most “talk” between the members of neurochemical ensembles happens outside of neurons, then neuroimaging studies of grey/white brain matter density and volume show only a part of the “story” of CBP biomarkers. Another part hidden from neuroimaging involves complex combinatorics and the diversity of receptor systems, including volume transmission. This hidden, processes-based side of biobehavioural regulation provides negotiations between neurochemical systems that largely affect what behaviour will be chosen.

Glutamate vs. GABA (GG) relations. The amount of excitatory glutamate (Glu) more than twice exceeds the amount of inhibitory Gamma-Amino-Butyric Acid (GABA) in the brain [127,129,130]. A massive amount of excitatory glutaminergic transmission induces those mass action in neurodynamics described by Freeman as the “Throw” trend, which are eventually organized in specific dynamical patterns (attractors), as the earlier selection within the “Catch” system. 

GG vs. MA relations. There are dramatic differences in functionality between Glu-GABA (let us notate this pair as GG), on the one hand, and other neurotransmitters, on the other hand. These differences align with the idea that, functionally speaking, the NTs are not there to do just excitation or inhibition but to assist a selection of DFs in behaviour. Here, are how MAs and ACh can be seen as a part of the “Catch” system, either modulating the massive amount of potential coming from the GG release or using GG as mediators: 

(A) In terms of the distributions of various NTs, the most striking difference relates to the sheer number of Glu- and GABA-releasing neurons versus MA-released neurons. As noted Glu and GABA are known for their dominant presence in the brain in comparison to any other neurotransmitter. In contrast to GABA, there are 1000 times fewer MA and ACh neurons than GABA-releasing neurons, and the ratio of MA-to-Glu neurons is even worse [131,132]. 

(B) GG neurons constitute the core structure of the cortex, with their pyramidal neurons having well-defined directionality and numerous synapses. Their activity creates most of the electricity that is picked up by EEG machines. In contrast, MA and ACh systems are more sneaky and almost structurally and electrically invisible. They have significantly fewer profound projections between the GG-produced giants; MA and ACh co-release their neurotransmitters with GG and often release their NTs into the extra-cellular space to act on all neurochemical ensembles without making classic synapses [38,39,40,41,127]. 

(C) GG NTs mostly work on local excitation and inhibition synapses, whereas MA and ACh systems have extensive branching, allowing them to connect distant areas for functional exchanges. 

(D) GG neurons are often organized in columns responding to very narrow features of stimuli. In the cortex, they generate associative waves of transmissions via horizontal, same-layer networks for processing contextual information. 

(E) GG neurotransmitters are only used by the brain, whereas MA and ACh neurotransmitters are used by both brain and body, and often more by the body than by the brain.

(F) A significant portion of components used to produce and decompose GG is provided by glial cells and not by neurons per se, whereas MA and ACh are composed and decomposed mainly by the chemicals of neurons. Glial cells, therefore, participate in waves generated duringneurotransmission. There is more complexity involved in the difference between neurotransmitters in terms of their composition, decomposition, locations, types of receptors and interactions with neuropeptides. These differences are not the subject of this paper but could be found in [36].

Ach vs. NE relations. There are differences in the distribution and structural complexity of the NE and ACh brain systems. Brain widespread NE systems were evidently linked to the orientational processes and attention to novelty [133,134], as well as HPA axis arousal when an immediate change in behavior is needed—supporting the idea of the “expansion” and “exploration” functionality of these systems [15,135,136,137]. In contrast to the extensive NE projections from the LC, the brain’s ACh systems have more distinct and relatively independent nuclei than any other NTs [138]. The ACh systems use different mechanisms and locations for each of the other neurotransmitters in organizing a behavioural response. The structural complexity and strategic positioning of receptors and projections allow the Ach system to be in charge of trimming multiple aspects of behaviour and so setting the chemical infrastructure for the “Catch” processes, namely:-orientational, via cholinergic cortical-basal forebrain projections and interactions between ACh and NE at that level;-integrational, via cholinergic interneurons regulating striatal DA-GABA networks; habit-formation, via the PPN-dLTA, nuclei, and-action maintenance (i.e., energetic) aspects via collaboration between ACh and 5-HT systems;-coordination of automatic selection of DFs by the cerebellum to the pANS level, under the close supervision of the ACh forebrain and lateral hypothalamic systems.-at the Autonomic Nervous System (ANS) level, the NE-based sympathetic ANS provides a fast and non-specific massive arousal (Throw), whereas the ACh-basedparasympathetic pANS acts selectively and is much more structured in its action. The pANS trims the DFs at the ANS levelcontrols the sANS activation of specific somatic functions. Such trimming of DFs, even at the very low level of behavioural regulation (including selective muscle contraction) provides precise control over locomotion [139].

Other examples of the T&C principle within the nervous system include an overproduction and then pruning of neurons around the birth in human infants [140,141,142,143] and volume (i.e., non-synaptic, extra-cellular) transmission mechanism that was documented for all neurotransmitter systems [120,121]. Figure 2 illustrates the contrast between behaviourism and the T& C principle supporting the constructivism approach (T&C principle). The behaviouristic script of information processing (still very common in neuroscience and public psychology) is that the nervous system processes all coming information and then sorts out the behavioural response of all DFs in behaviour, theoretically available to a phenotypic individual representing a given biological species (Figure 2A). Constructivism and the T&C principle suggest that nervous systems constantly generate (construct) dynamical patterns (including cellular- and inter-cellular levels cycles) and likely just ignore most incoming information unless it is relevant to the individual. The nervous system works on a small, most relevant set of DFs in perception and behaviour based on the executive capacities of the given individual. In the presence of uncertainty (whether in the orientation of behaviour or during a choice between several scripts of actions), various levels of neuronal regulation use the T&C “to shed light on the mess” and make the contrast between the elements under review more clear (Figure 2B). 

Current neurochemical research often focuses on single spots in the brain releasing specific NTs or having high receptor density. It might be beneficial to investigate the chain of several neurochemical “relays” within the nervous system along the functional stages that the constructivism approach describes. These relays would be based on the T&C mechanisms, in which the Throw part is associated with the releasing “spray” of the chemical agents and the Catch part—with the strategic location of receptors that could be activated during the specific and not all aspects of action construction. Attempts to map receptor density indeed showed that receptors binding neurotransmitters and opioid peptides are strategically located in specific brain structures [10,144,145,146,147,148]. The reported differences in the functionality of brain structures can be, therefore, associated with the differences in receptor locations, i.e., differential neurochemical composition of these structures. Figure 1 gives an example of some brain structures differentially involved in the processing of degrees of freedom and composition of sustainable behaviour. 

A proposed earlier neurochemical framework that is briefly described in the next section can be used as a set of hypotheses for such relays.

## 5. Structuring the Setup: A 12-Component Framework of Universal Aspects of Constructive Processes

### 5.1. A Neurochemical Framework Functional Ensemble of Temperament Uses a Constructivism-Based Classification

The outcomes of many current projects are often presented as connectivity maps listing numerous excitatory-inhibitory associations. Such outcomes generate the “big data problem”, when researchers do not know how to make sense of it. They often count on blind statistical software (such as factor analysis or data-mining [1]) to find new useful insights. Theory-based hypotheses often help to increase the efficiency of data collection and analysis, and the classificational principle of Functional Constructivism offers a set of such hypotheses. *Functional Constructivism* principle links the functional roles of neurochemical systems to the universal stages and functional aspects of the construction of behaviour. The neurochemical framework, Functional Ensemble of Temperament (FET) [7,8,14,16,17,18], based on this classification principle, summarizes experimental evidence from neurochemistry, differential psychology, psychiatry, endocrinology, and addiction research and represents CBPs as the product of contingent interactions among specific neurotransmitters, opioids and hormonal systems (Figure 1 and Figure 3). Our review of functional neurochemistry showed that, despite the diversity of receptors’ actions within each neurotransmitter system, there is overall a strong correspondence between the roles of these systems and universal stages and functional aspects of behavioural regulation. The FET has a 12-block structure aligned with the three main aspects of action construction: expansion of degrees of freedom (orientation), selection/integration of a program of action, and energetic maintenance (including decomposition of unneeded degrees of freedom) (three columns of Figure 3). 

Moreover, the functions of neurochemical systems reviewed in the FET are viewed in association with three types of cycles related to phenotypes functioning (three top rows of Figure 3): -within-body cycles (“physical”), regulated by 5-HT, NPs and gut microbiota systems, with a prominent role of hypothalamic—anterior pituitary peptides and hormones such as Somatostatin, Growth Hormone and the status of the thyroid system. These cycles also include several levels of sub-cycles for the maintenance of neuronal activities, for example, in the form of neuropils [75,149]. Neuropils are known as glia-neuronal complexes taking spaces between cell bodies and including dendrites, axons, synapses, and microvasculature. Since we discuss here primarily the “managerial” neurochemical systems, which impact emerges at the behavioural level, there is no space here to discuss the numerous proteins and enzymes regulating the neuropils complexes at the cellular level;-cycles that include interactions with other bodies (peers, offspring, prey and predators) (“social”) regulated by other hypothalamic-pituitary systems (prolactin) and the “social” hormones oxytocin and vasopressin released from the posterior pituitary. Behavioural regulation at this level goes much beyond the communicative functions as involvement in these cycles allows sharing material (such as closing, housing, food sources, transport) and informational (knowledge, motivations, attitudes) infrastructures;-cycles that include the tuning of behaviour to more extensive infrastructures that might not be immediately present, including probabilistic features of reality (“probabilistic”). To ensure the maintenance of probabilistic activities, many structures and chemical systems of the brain get involved, with glial cells playing a major role.

Moreover, there are genotypes and sex-reproduction cycles regulating human behaviour at neurochemical, anatomic, social and probabilistic levels, and, therefore, controlling all phenotypic cycles. 

As suggested in previous papers [7,19], behaviour does not start from orientation (and from stimuli) but serves the maintenance of the listed four cycles. The more discrepancy between needs and capacities for maintenance of these cycles, the more integration of behaviour is launched (Figure 3). If a simple re-integration of learned DFs is not sufficient (such as in complex, uncertain or novel situations), then the orientational systems are getting involved to a higher degree. Here, we have to underline that integration- and orientation-related systems are almost always involved to some degree to ensure the adequacy of actions, even for the most automatic actions. Our several earlier reviews highlighted the research of Trevor Robbins and colleagues related to the roles of DA-5-HT systems in behavioural integration (plasticity) and its deficiencies in impulsivity [12,13]. DA systems work closely with ACh and GABA in the integration and programming of behaviour [150,151]. With the choice of more established (automatic) units of behaviour, the role of delta opioid receptors in basal ganglia also becomes crucial [8,11,12,13,15]. There are distinct neurochemical and neuroanatomic differences in the regulation of the following types of integration of behaviour:-immediate integration, triggered by environmental stimuli without plans of actions and mainly regulated by the autonomic nervous system with very limited involvement of orientation systems (known as impulsive, premature initiation of actions).-automatic integration of actions and cognition with a developed program that sequences behavioural elements and-novel integration of behaviour when a new program (choice and sequencing of actions) is required—common in complex and uncertain situations.

Three types of behavioural orientation also relate to two more deterministic aspects of behaviour—orientation to sensations (physical, body-oriented type) and to expectations of other people, empathy (i.e., social type) [18]. The probabilistic type of orientation is known as contextual information processing and learning abilities about not immediately present events [17] (Figure 3).

Finally, the FET framework summarized the functionality of three opioid receptor systems as inducing dispositional emotionality or premature integration aside from normal selection processes described above [8,16,18] (bottom row of Figure 3). The kappa opioid receptors system (KOR), coupled with the NE and Glu release, is seen as a limbic amplifier of sensory-orientational mobilization; the mu-opioid receptors (MOR), coupled with 5-HT and DA, are considered within the FET as an amplifier of approval of alternatives. The third, delta-opioid receptor system (DOR), is coupled with DA and GG networks in the basal ganglia and often creates heterodimers with MOR, working in one direction and binding each other’s peptides. DOR can be presented as a system assisting the further selection and initiation of actions. Activation of these opioid receptors and possibly high presence of peptides binding to them can generate at least three emotional dispositions in behaviour that occur regardless of events, i.e., Neuroticism, dispositional Satisfaction and Impulsivity [8,16,18] (Figure 3). 

The structure of the neurochemical framework FET highlights the universal architecture of action construction, so it applies to any behaviour, no matter how dysfunctional. The FET, therefore, supports the idea of a health-clinical continuum [2,3,4,5,6], unifying healthy neurochemical regulation (seen in temperament traits) and various degrees of dysregulation, including the clinical spectrum seen in psychopathology. A potential correspondence of the FET components to classic symptoms of all major psychiatric disorders was presented in our earlier reviews [7,8,19]. The FET links every psychiatric symptom or temperament (bio-behavioural) trait to a team of neurochemical systems (the *multimarker approach)* and suggests that there is no one-to-one correspondence between specific behavioural patterns and any single neurotransmitter, hormone or neuropeptide [7,8,14,15,16,17,18]. There is a natural partial overlap between these teams because the FET components reflect the stages in a progression of the construction of actions, and so the transition between stages requires team’s mutual regulation. 

The FET structure appeared to be fruitful for neurophysiological [152,153,154], developmental [155], and clinical [3,4,6,155,156] investigations. Temperament profiles of clients with psychiatric disorders followed the predictions of the FET model when investigated with the FET-related test and Beck Anxiety Inventory [6]; the Five Factor Inventory of personality [157]; Zuckerman’s Sensation Seeking Scale [157]; the Hamilton Depression Inventory [4,6]; Personality Assessment Inventory [156]; documented symptoms of Generalized Anxiety [3], Major Depression [6], psychotic symptoms ([155,156] and comorbid anxiety and depression [6]. 

All CBPs could be presented as particular holographs in 12-dimensional space of the constructivism components outlined in the FET. Instead of correlating very diverse actions of individuals from various cultures, constructivism suggests that it is not the content of the CBPs but their dynamics (speed, range, sustainability) that could be individually consistent. The FET outlines the components of dynamical styles universally generating the CBPs (healthy temperament traits or symptoms of psychopathology), no matter how diverse individual actions can be. The differences in the CBP dynamics involve the behavioural construction styles contrasted by their novelty, urgency, complexity, level of automatism, social networking, speed of change of the situation and physical demands.

The same 12 aspects could be seen in the classifications of contexts (and SEP types), i.e., higher level of behavioural regulation that contains a very important diagnostic and predictive information about the individual’s SEP infrastructures. The same universal dynamical aspects of construction could be used in the analysis of neuronal activity (maintenance of potentials, their variability, the ease of their composition, “social” mass action and networking between neurons and the use of various amplifiers). Correspondence between the categories for SEPs, behavioural or neuronal run-aways, start-ups and CBPs analysis using the same FET components makes these blocks of information compatible for tracing their interactions. Thus, SEP infrastructures have differences in terms of demands for energetic physical capacities, sociability or sustained attention, and individual preferences between SEPs having these demands signal bio-behavioural markers of SEPs. Individuals preferring athletic activities, buildings or garden constructions would respond to the first type of demands; individuals capable of prolonged communications would gravitate to social networking and participation in parties, whereas individuals gravitating to a third type of SEP would like to join chess or reading clubs. After years of activities within their SEPs, people’s perception becomes tuned to specific reinforcers, objects, concepts, and societal and physical infrastructure, securing their SEP “bubbles” and establishing particular patterns in their neuronal processes. The presentation of each SEP, run-aways and start-ups in the degree of expression along all 12 FET components could highlight the specificity of the biomarkers and environmental markers of CBPs, facilitating the development of their taxonomies.

Having a universal structure of constructive processes for multiple levels of behavioural regulation helps to trace “diagonal” effects in interactions between levels [49] and potentially improves the efficiency of studies in behavioural genetics. In fact, diagonal effects are known in the impact of the specifics of human activities and societal infrastructure on the development of the human brain, including the structure of neuropils in the frontal lobe, especially the prefrontal cortex [149]. 

The structuring using 12-component and their contingent relationships could be, therefore, used in the methodology of the H-project related to neurochemical, genetic and environmental factors of CBPs.

### 5.2. Possible Neurochemical and Genetic Investigations within the H-Project Should Target the T&C “Relays”

The 12 aspects of behavioural regulation that the FET uses to categorize the functional specialization within neurochemical systems can be used as a framework for structuring neurochemical investigations. Currently, there is a puzzle of a mismatch of the locations releasing hormones, slow neuropeptides or neurotransmitters and the locations of receptors that bind these chemical agents [10]. In the proposed T&C scenario, nervous systems throw substances and actions, similar to a child, first making a sand pile and then using the forms to construct preferred shapes from the sand. The placement of receptors by the nervous system likely follows its body-capacities/needs-, habits-, history-, and context-related biases. The biases determining what information to look for and what actions toprefer, allow nervous systems to not perceive the majority of surrounding stimuli but to have a limited, pro-active search for the relevant information. During such a pro-active, SEP-biased search, a small portion of stimuli that is chosen for perception receives “the royal treatment” in the form of welcoming mass action of neurons generating anticipatory attractors in neurodynamics described by Freeman. The ability of the nervous system to direct its perception by “Throwing” resources to the relevant aspects of the environment facilitates the process of matching the compatibility between the state of the needs/capacities of the individual and the environmental offers.

In analysing these T&C relays, special attention could be given to the contingent relationships between the NT systems “tossing” the behavioural regulation between each stage of behavioural construction. Figure 3 lists these neurochemical systems with an assignment to functional aspects of behavioural regulation. The neurochemical investigations of the T&C relays can have a look at the mutual regulation of the neurochemical systems within each FET component and between leading neurochemical systems of several components. This systematic approach will at least save resources by having a structured setup for investigations. The potential relays-analysis, for example, can compare constructivism and “reactivism” hypotheses. In constructivism, behavioural regulation starts from cycles if there are strong maintenance systems suppressing run-away and start-up processes (associated in FET with 5-HT, ACh, NP, cortisol and MOR systems). The established SEP-individual-neuronal “diagonal” within these multi-level cycles would generate specific T&C relays sensitive mostly to the content of these cycles. If the Reactivism hypothesis is correct, then the neuronal ensembles and, therefore T&C arrangements of releasing and receptor sites of NT systems would show a non-specific diversity of locations of these sites. 

As another example, the relay between the integration of a program of actions and orientation could be analyzed as an analysis of the directionality of the orientational “Throw”-release within the Glu and NE systems and their “Catch” structure within the systems regulating integrative aspects of behaviour (i.e., DA-ACh networks). As noted earlier, constructivism highlights the three types of integration (plasticity, which depends on orientation the most; impulsivity and automatic integration, which depends on orientation the least) [7,8,49], each having their specificity in using the DA-GABA-Ach networks. The FET offers several other verifiable hypotheses about the directionality of the neurochemical T&C relays in release sites and receptor density associated with each of the three scripts of behavioural construction [8]).

Such a structured approach using a neurochemical framework might be more efficient in genetic investigations than behavioural genetics studies using lexical personality models [158,159]. Instead of using genetic correlates of traits based on social constructs, this approach would look for genetic correlates of interactions between specific neurochemical functional blocks of behavioural construction. After all, genes are a clear example of constructivism and also represent biochemical systems of behavioural regulation. Using temperament models based on the neurochemical framework in genetic studies could be a promising setup for the H-project and other investigations.

## 6. Conclusions

There is a need for an international project, “Hippocrates”, that would launch more systematic investigations of the roles of internal neurochemical and environmental factors in consistent behavioural patterns. This paper reviewed several constructivist principles and their applicability to a potential H-project. The paradoxical principle of constructivism suggests that there is no repetition during repetitions or similar actions (processes). Instead, neural and behavioural processes use a multi-level selection of degrees of freedom (Figure 1) complemented by the generation of intermediate local loops of “run-away” and “start-up” integrations. Constructivism is confirmed at the molecular level, at the level of neurotransmission, at the level of networks regulating repeated actions, and at the level of behaviour. Here, we pointed to three methodological aspects for the H-project’ setup, in which constructivism paradigm differs from the currently dominating approaches.
(1)Where to measure: Investigations of neurochemical biomarkers of CBPs should be the main aim of the H-project. However, this review pointed out that the biomarkers of the CBPs are not just in the brain (that is the main focus of the Connectome-like projects) but also in endocrine and microbiome systems. There should be, therefore (complementary to neurochemical investigations of T&C relays), regional comparisons of the environmental factors [2] (such as stresses, diets, exposure to common psychostimulants and toxins) influencing these systems. Moreover, the review underlined the importance of tracing Specialized Extended Phenotypes associated with specific CBPs.SEP relates to the environmental establishments created or reinforced by people during their ontogenesis based on their bio-behavioural regulatory preferences and their CBPs involving these establishments. Then, in turn, these individualized social, physical and informational infrastructures (such as social relations including pets, people’s IDs, professional history, and outcomes of their physical actions) regulate people’s everyday life and, therefore, associated CBPs.(2)What to measure: many current projects inherit a reactivity paradigm known in behaviourism. In such a reactivity paradigm, experimental conditions are considered to be the leading factors of the differences in neuronal and behavioural variations: experimental events are induced, and the brain reacts as if there was no brain activity before these events. In contrast to that, constructivism points to the pro-active nature of CBP biomarkers seen in anticipatory neurodynamics described by Walter Freeman, neuroendocrinal regulation and in the Throw&Catch phenomena, largely unrelated to the current events. The T&C, seen at the multiple levels of neuronal and neurochemical regulation, self-generates an excessive variance in some neuronal subsystems, to be pruned by other subsystems in a relay manner. At each stage of the T&C, a selection of DFs becomes tighter, with specific Throw and Catch subsystems for each stage. Measurements, therefore, should trace key aspects and stages of behavioural construction:
In the H-project, brain neurochemstry studies should measure not only NT relays but also endocrine variables indicative of the individual’s needs and capacities.The positioning of NT-releasing sites and their receptors is neither even nor random in the brain. Instead, it follows the constructivism trend, with the Throw&Catch relays aimed to highlight the most relevant and suppress irrelevant DFs in behaviour. The investigations of neurochemical systems, therefore, could be organized in a more systematic manner, following the relays between the NT release and binding sites according to the verifiable constructivist hypotheses about these relays. Such targeted measurements of these sites tracing several relays can be more informative than the current focus on one single site for the NT release or receptors density. These neurochemical investigations should mind regional comparisons of the environmental factors influencing the CBPs [2] (as noted above). This review suggested measuring behavioural and neuronal events that support CBPs but by themselves are not consistent: “start-ups” (initiated activities) and “run-aways” (incompleted activities). For example, current studies of CBP biomarkers identify CBPs mostly using self-reports, including clinical interviews and questionnaires asking about the most frequent and consistent events/experiences. This review pointed out the potential informational value of the structured analysis of the personal and professional, indicative of background and transient processes that led to the consistent CBPs. At the neuronal level, studies in neuroscience often focus on the most visible neuronal “hardware” (brain connectivity, activation of brain regions) within an individual’s nervous system and trace their associations with CBPs. Here, we suggest conducting not the “hardware” comparisons but the comparisons of the components of the neurochemical T&C relays involved in the construction of actions for individuals with different SEP and CBPs.(3)Structuring the setup: The outcomes of many current projects are often presented as connectivity maps listing excitatory-inhibitory associations. Since these associations are numerous, researchers often face a “big data problem”, not knowing how to make sense of it and counting on blind statistical software (such as factor analysis or data-mining [1]) to help with new useful insights. Theory-based hypotheses often help to increase the efficiency of data collection and analysis, and the principle of Functional Constructivism offers a set of such hypotheses. This principle points to the universality of dynamical features and stages of behavioural construction and suggests using these stages as the structural design for the outcomes of the H-project. Based on this principle, the neurochemical framework Functional Ensemble of Temperament highlights the correspondence between the functionality of families of neurochemical systems and 12 universal aspects of behavioural regulation. These aspects relate to orientation, integration of behaviour and maintenance of specific cycles of individual survival assessed separately for physical (body), social (other bodies) and probabilistic aspects of behaviour. Three other FET components relate to dispositional emotionality and the HPA-driven integration of behaviour(Figure 3). The FET is a conservative general summary of functional specialization within neurotransmitter systems; however, much more work is ahead to complement this summary with the details on receptor functionality within each of these systems. The FET offers verifiable hypotheses about the neurochemical T&C relays [8]). Moreover, the FET structure can be used in experimental studies of environmental “bubbles” (Specialized Extended Phenotypes) that support an individual’s CBPs. [7])

These are a few thoughts towards a much-needed international neurochemical H-project. Much more analytic and theoretical work should be done before the call for sponsors of such project due to the complexity and invasive nature of neurochemical and endocrine investigations [1,2,160]) This project should be multi-disciplinary, including not only neurochemical studies but having cultural samples contrasted by the specific components of diets (such as tryptophan, soy, probiotic products, meats, and caffeine), climates (such as humidity and amount of sunlight), exposure to toxins (including air and water pollution) and common psychostimulants (such as alcohol, nicotine opioids and cannabis). Discussions of a need for a multidisciplinary international project have already started among academics [2], but more organizational steps are needed to move this project forward. Similar to the international project CERN devoted to high-energy physics, these organizational steps should go hand in hand with the analysis of the scientific background of this project, influencing the design of variables and research questions. Moreover, the regional comparisons of diets, climates, SEP types, etc. complementing neurochemical investigations are possible only with the cooperation of multiple scientific and public communities representing various counties. Therefore, similarly to the CERN project in high-energy physics, the H-project could be developed only with a setup of an international Task Force and extensive discussions at many levels of scientific and international organizations. 

To prepare this review, the author used PubMed, CrossRef, APA PsychInfo databases and a targeted search using the keywords related to specific neurochemical systems, constructivism and other concepts of neuroscience.

## Figures and Tables

**Figure 1 brainsci-13-00039-f001:**
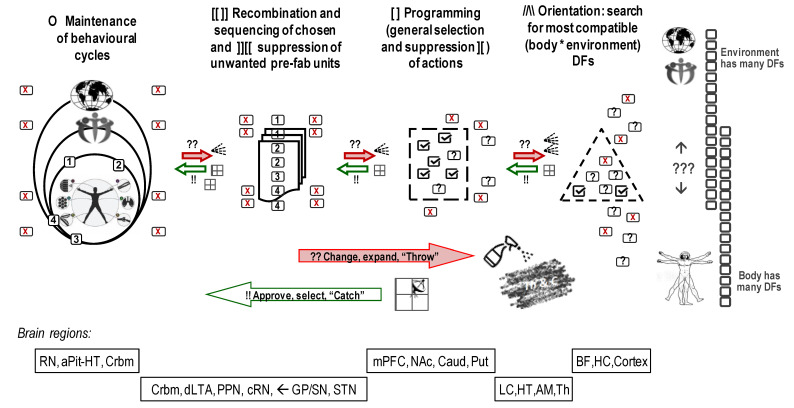
Action as a result of the selection of degrees of freedom (DF). This Figure uses symbols as formal notations for behavioural aspects. Constructivism suggests that behaviour starts from the maintenance of current cycles (denoted as O), and if the current behavioural alternatives do not fit the needs for cycle maintenance, behavioural regulation launches the re-integration of prefab (previously learned) actions (marked as [[]]). If prefab elements are insufficient, an integration of a new program of actions starts (marked as []), and if a program cannot be composed, then orientation processes increase (denoted as //\\). All these blocks are active all the time but to a different degree that depends on the complexity, novelty and uncertainty of situations and capacities of the individual. The lower level highlights the brain structures involved in the relevant stages of action construction: RN –raphe nucleus; mPFC—medial) prefrontal cortex; HT—hypothalamus; HC—hippocampus; BF—basal forebrain; LC—locus coeruleus; (dL, V)TA—(dorsolateral, ventral) tegmental area; Th—thalamus; AM—amygdala; NAc—nucleus accumbens; Caud—caudate nucleus in ventral striatum; Put—putamen; STN—subthalamic nucleus; PPN—pedunculopontine nucleus; Crbm—cerebellum; GP—globus pallidium; SN—substantia nigra.

**Figure 2 brainsci-13-00039-f002:**
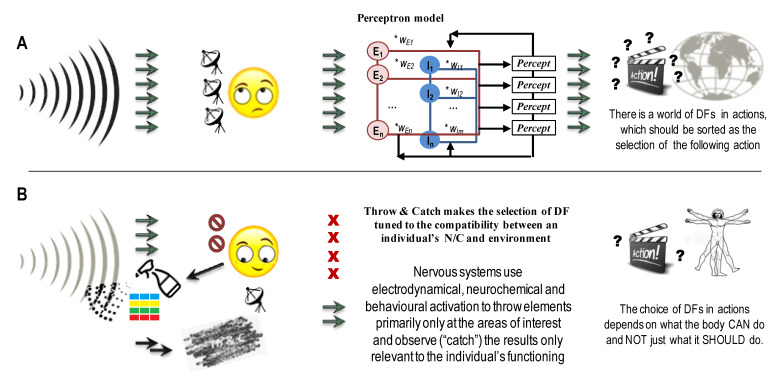
The contrast between the Perceptron and classic excitation-inhibition models (**A**) and the Throw & Catch (T&C) principle (**B**). (**A**): In common Perceptron-like models, perception is viewed as receptors (depicted as antennas) passively receiving and processing ALL surrounding information using sets of inhibitory and excitatory networks. Biases are given as weights added to the size of the signals. (**B**): In the T&C principle, the nervous system throws substances and actions towards relevant targets, similarly to a pencil technique looking for the dents on a paper sheet. The “Throw” processes proactively generate an additional variance to increase the visibility of details needed to select degrees of freedom. The “Catch” processes represent much more organized and complex parts of the nervous system compared to the “Throw” processes. The Catch systems use strategic placement of receptors (depicted as an antenna) to analyze the resulting “dents”. The directionality of “Throw” and the position of Catch follow the body- and experience-related biases. The coloured rectangular logo symbolizes the role of neurochemical systems in inducing these biases. Using T&C as a self-regulated “flashlight” to amplify the details of perceiving DFs allows nervous systems to save resources by not sensing the majority of surrounding stimuli and have a proactive search for the relevant information.

**Figure 3 brainsci-13-00039-f003:**
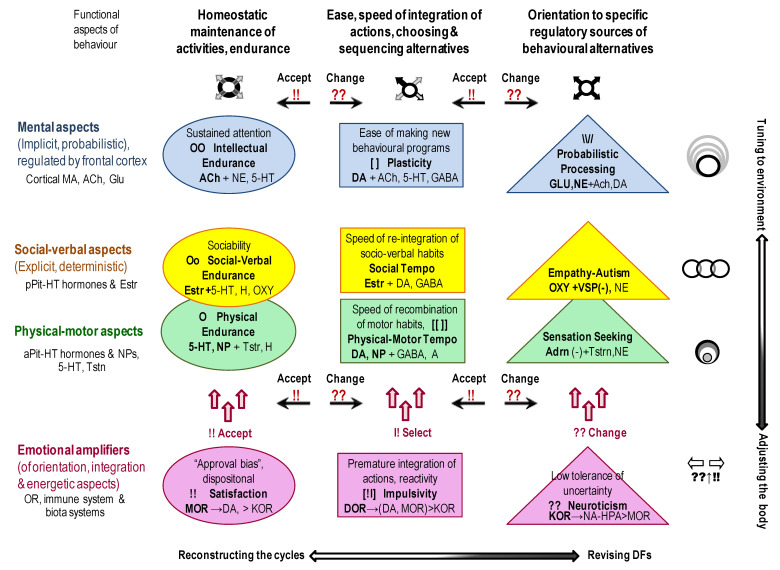
Neurochemical framework Functional Ensemble of Temperament (FET) summarizing the action of neurochemical systems in the regulation of 12 universal aspects of action construction. Red “?” and “!!” signs symbolize emotional amplifiers based on opioid receptor systems, as described in our earlier reviews. When selected DFs do not work, a further search for alternatives is increased (“?”), and if it does—the chosen alternatives are approved (“!!”). The “?” system that generates the release of the catecholamines (DA in familiar elements of behaviour and NE in novelty), as well as the HPA axis arousal in rough emotional response, is represented by KOR, some microbiota and cytokines systems. The “!!” system that generates the release of DA and suppresses the HPA axis activation is based on the action of MOR (in the ventral striatum), DOR (in dorsal striatum) and some microbiota. Note: ACh: acetylcholine; NE: noradrenaline; 5-HT: serotonin; DA: dopamine; OXY: oxytocin, VSP: vasopressin, Tstr: testosterone; Adr: adrenalin (and its deficient cycles); GC—glucocorticoids (including cortisol dysregulation); ORE: orexins; NP: neuropeptides; Glu—glutamate; GG—Gamma-Aminobutyric Acid and Glu; (M/D/K)OR: (mu/delta/kappa) opioid receptor systems; ANS—autonomic nervous system.

## Data Availability

Not applicable.

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
