# Peer review of "Analytic Background in the Neuroscience of the Potential Project “Hippocrates”"

_brainsci, 2022, doi:10.3390/brainsci13010039_

Round 1
Reviewer 1 Report
The Author produced a well-written article concerning the principles identified in analytic neuroscience that could be used in the setup of an international project “Hippocrates” (H-project). This project could investigate the roles of internal neurochemical and environmental factors in consistent behavioral patterns. The number of references cited is significant and the reference list covers the relevant literature adequately. However, I have some minor comments, as below:
1. The reviewer has a problem with qualifying this article as the Review because it is more similar to the introduction to the grant application than to the review article. Thus, the author should specify what type of the review this article represents.
2. Other articles by the author are over-cited in the reference list and the manuscript text. The author is asked to limit self-citation to the most important papers related to the subject of the manuscript.
3. Since many of the earlier own papers cited by the author concern a similar topic, please indicate the novelty in the current manuscript.
4. I found a problem with the manuscript page numbering in the manuscript file. Please check and correct the page numbering.
Author Response
Reviewer 1
- The reviewer has a problem with qualifying this article as the Review because it is more similar to the introduction to the grant application than to the review article. Thus, the author should specify what type of the review this article represents.
A comment was added to p.4 of the ms
- Other articles by the author are over-cited in the reference list and the manuscript text. The author is asked to limit self-citation to the most important papers related to the subject of the manuscript.
Six self-citations were removed, and currently, the author’s citations represent less than 10% of all citations (this is within the normally allowed limit). The remaining citations are needed to complement the provided references with sources describing the earlier developed neurochemical framework published as a series of 6 papers. The proposed framework described the functionality of multiple neurochemical systems in regulating 12 behavioural aspects, and this complexity required a division of the discussed material into smaller topics (accommodated in 6 papers). Repetition of references from those 6 papers would increase the manuscript’s volume.
- Since many of the earlier own papers cited by the author concern a similar topic, please indicate the novelty in the current manuscript.
The novelty of this ms is its primary focus on the principles of a setup of a possible international project. Other papers of this author discussed the functionality of multiple neurochemical systems in regulating of 12 behavioural aspects, or experimental studies using the proposed framework, or presented a review pointing to systemic tendency in sex differences documented in psychological abilities and disabilities, or outlined the concepts of diagonal evolution and Specialized Extended Phenotype.
- I found a problem with the manuscript page numbering in the manuscript file. Please check and correct the page numbering.
I used the journal’s template and was not able to change pagination in the places where the Figures should be inserted. Perhaps the journal can advise me on what to do.
Reviewer 2 Report
This work is an analytical review of the literature, in which the data substantiating participation in the Hippocratic project are presented in great detail and verbosely. The main problem of this work is its verbosity and the availability of data that are well known and are included in the curriculum of medical universities. It is necessary to conduct a critical revision of this work and shorten it, making it more specific and meaningful. In this version, the article is very long. This review contains a lot of data that are not necessary for understanding the essence of the issue, for example, the information “Implicit opposition between behaviorism and constructivism in neuroscience” or “Plurality of candidates” looks completely superfluous, supports “repetition without repetition” at many levels of behavior regulation, but leads to the problem of degrees of freedom. These sections contain a lot of generalized and unnecessary information from the field of the history of medicine. Thus, this work needs to be revised, in particular, to significantly reduce some sections and remove well-known medical information from this work.
Author Response
Reviewer 2
The main problem of this work is its verbosity and the availability of data that are well known and are included in the curriculum of medical universities. It is necessary to conduct a critical revision of this work and shorten it, making it more specific and meaningful… This review contains a lot of data that are not necessary for understanding the essence of the issue, for example, the information “Implicit opposition between behaviorism and constructivism in neuroscience” or “Plurality of candidates” looks completely superfluous, supports “repetition without repetition” at many levels of behavior regulation, but leads to the problem of degrees of freedom. These sections contain a lot of generalized and unnecessary information from the field of the history of medicine.
The section “Implicit opposition between behaviourism and constructivism in neuroscience” was removed, and the text was slightly trimmed. However, the “Multiplicity of candidates” section remained as it is one of the key explanations of how the problem of degrees of freedom can be solved. The author disagrees that the material in this section is taken from the field of the history of medicine. The DF problem is largely unknown in medicine, and the provided material uses mainly references from neuroscience and not medicine (in line with the title of the Brain Sciences journal).
Reviewer 3 Report
This is a review article about an interesting subject, which is the assessment of several constructivist principles and their applicability for a potential project “Hippocrates”, aimed to investigate the roles of internal neurochemical and environmental factors in consistent behavioral patterns. The manuscript is generally well structured.
English language and style are fine but there are some minor issues that need to be addressed.
The author has performed a very extensive literature search. Perhaps, a small paragraph describing the methodology used during this procedure (for instance which terms and which databases were used during literature search etc.) would further improve the quality of this manuscript.
To my opinion, conclusions are written in a detailed and critical manner, making the key points comprehensible to the readers.
Author Response
Reviewer 3
This is a review article about an interesting subject, which is the assessment of several constructivist principles and their applicability for a potential project “Hippocrates”, aimed to investigate the roles of internal neurochemical and environmental factors in consistent behavioral patterns. The manuscript is generally well structured. English language and style are fine but there are some minor issues that need to be addressed.
Thank you for these kind words.
The author has performed a very extensive literature search. Perhaps, a small paragraph describing the methodology used during this procedure (for instance which terms and which databases were used during literature search etc.) would further improve the quality of this manuscript.
The sentences about databases were added to the paper's second and last paragraph of the text.
To my opinion, conclusions are written in a detailed and critical manner, making the key points comprehensible to the readers.
Thank you for these kind words
Round 2
Reviewer 2 Report
This work contains a lot of unstructured information. The article is difficult to understand and contains a lot of raw information, which gives the impression of insufficient study of the topic by the author. This review provides a lot of descriptive and unproven information as examples and lacks the necessary evidence base. The author did not sufficiently revise and make the recommended corrections to the new version of the work.
Author Response
I disagree that the paper did not provide the needed evidence, but I agree that the previous ms needed structuring. The text was edited to bring the take-home message to the beginning and to highlight the conclusions of the paper.